# Explaining machine-learning models for gamma-ray detection and identification

Mark S. Bandstra[1]*, Joseph C. Curtis[1], James M. Ghawaly Jr[2], A. Chandler Jones[1], Tenzing H. Y. Joshi[1]

**1** Nuclear Science Division, Lawrence Berkeley National Laboratory, Berkeley, California, United States of America, **2** Physics Division, Oak Ridge National Laboratory, Oak Ridge, Tennessee, United States of America

* msbandstra@lbl.gov

## Abstract

As more complex predictive models are used for gamma-ray spectral analysis, methods are needed to probe and understand their predictions and behavior. Recent work has begun to bring the latest techniques from the field of Explainable Artificial Intelligence (XAI) into the applications of gamma-ray spectroscopy, including the introduction of gradient-based methods like saliency mapping and Gradient-weighted Class Activation Mapping (Grad-CAM), and black box methods like Local Interpretable Model-agnostic Explanations (LIME) and SHapley Additive exPlanations (SHAP). In addition, new sources of synthetic radiological data are becoming available, and these new data sets present opportunities to train models using more data than ever before. In this work, we use a neural network model trained on synthetic NaI(Tl) urban search data to compare some of these explanation methods and identify modifications that need to be applied to adapt the methods to gamma-ray spectral data. We find that the black box methods LIME and SHAP are especially accurate in their results, and recommend SHAP since it requires little hyperparameter tuning. We also propose and demonstrate a technique for generating counterfactual explanations using orthogonal projections of LIME and SHAP explanations.

## Introduction

The analysis of gamma-ray spectra is an important task in the realm of radiological and nuclear security [1, 2]. Gamma-ray spectra can be complex, due to the many possible sources of natural and artificial backgrounds, the large variety of radioactive isotopes and special nuclear material (SNM) that may be relevant to a particular problem, and the possibility of instrumental effects such as calibration drift. To deal with these complexities, machine-learning methods, especially artificial neural networks (ANNs), have been used in gamma-ray spectroscopy for over three decades [3–8].

Recently there has been an increase in the use of machine learning to analyze an entire gamma-ray spectrum, often to determine the presence of certain isotopes and quantify their relative strengths [9–24]. As more complex approaches are used for the detection and quantification of isotopes using gamma-ray spectroscopy, and especially as those approaches are

**Data Availability Statement:** The neural network model and code to generate and plot explanations are available at the following code repository: https://gitlab.com/lbl-anp/radai/explanations The

training, validation, and testing data are available at the following DOI: https://doi.org/10.7941/D1XC97.

**Funding:** This work was performed under the auspices of the U.S. Department of Energy by Lawrence Berkeley National Laboratory (LBNL) under Contract DE-AC02-05CH11231. The project was funded by the U.S. Department of Energy, National Nuclear Security Administration, Office of Defense Nuclear Nonproliferation Research and Development. This manuscript has been authored in part by UT-Battelle, LLC, under contract DE-AC05-00OR22725 with the US Department of Energy (DOE). The US government retains and the publisher, by accepting the article for publication, acknowledges that the US government retains a nonexclusive, paid-up, irrevocable, worldwide license to publish or reproduce the published form of this manuscript, or allow others to do so, for US government purposes. DOE will provide public access to these results of federally sponsored research in accordance with the DOE Public Access Plan (http://energy.gov/downloads/doe-public-access-plan). The funders had no role in study design, data collection and analysis, decision to publish, or preparation of the manuscript.

**Competing interests:** The authors have declared that no competing interests exist.

potentially encountered in high-stakes security applications, it will become increasingly necessary for researchers and end-users to understand how algorithms are reaching their conclusions [25]. Such considerations in the wider artificial intelligence community have led to the burgeoning field of explainable artificial intelligence (XAI) [26, 27], which addresses the need to increase transparency and user trust in machine-learning algorithms. Explanations in the gamma-ray spectral domain will enable end users to not only have trust in the results, but also to more efficiently understand and adjudicate alarms during field operations.

Some researchers have taken note and made the first attempts to apply XAI techniques to the outputs of these new machine-learning approaches [18–20]. In reference [18] the authors apply saliency mapping [28], which gives a clear result for the activations of $^{137}$Cs, namely that the 662 keV region is most important in identifying the presence of that isotope. However, they found that saliency sometimes failed to provide a useful explanation, such as when it was applied to an $^{241}$Am spectrum, where saliency showed no activation with respect to the strong 60 keV line from the isotope. In reference [19], SHapley Additive exPlanations (SHAP) [29] was used to examine the results of models that determine uranium enrichment levels from high purity germanium (HPGe) detector spectra. SHAP was able to indicate the correct peak regions corresponding to the different uranium isotopes and the strength and direction of their influence on the enrichment result, although they were not clear about the details of how SHAP was calculated for their data, such as what type of scheme was used to mask out or disable spectral features. Reference [20] implemented and compared several explanation methods for use with low resolution plastic scintillator spectra. The gradient-based methods of Gradient-weighted Class Activation Mapping (Grad-CAM) [30], Integrated Gradients (IG) [31], and Layer-wise Relevance Propagation (LRP) [32, 33] are compared to the black-box methods Local Interpretable Model-agnostic Explanations (LIME) [34] and SHAP [29]. Their conclusion was that Grad-CAM and LIME give unusable results, IG and LRP are possibly useful, and SHAP gives results that seem highly relevant and intuitive. However, since LIME and SHAP are closely related methods, it is unlikely that their performance would be so different. With few details about the implementations used for LIME and SHAP, it is difficult to infer the cause of this discrepancy. Taken together, the conclusions so far are that some approaches appear to yield better and more intuitive results than others, and that the best methods indicate that the models generally use the isotope-specific photopeak regions (or Compton-edge regions in the case of the plastic scintillator detectors in reference [20]) to determine the model outputs. However, there is also a need for a deeper understanding and articulation of the application of these methods to spectral data, particularly in cases where the gamma-ray spectra are more complex as is the case for most nuclear material.

In this work we focus on the problem of spectral detection and identification in urban search using simulated NaI(Tl) detectors, which are considered to have moderate energy resolution. We implement some of the explanation methods that have been explored by others and find that in some cases modifications are needed to more fully adapt those methods for gamma-ray spectral data (Explanation methods). These modifications have not been noted in other work and may account for the mixed results seen across the methods. We also propose a method for producing counterfactual explanations, i.e., why the model predicted one class over another. To demonstrate these methods, we use an ANN model that is consistent with recent literature and trained on the latest in synthetic data for urban search (Data and model). Finally, we show the results of using these explanation methods (Results) and discuss their implications (Discussion), specifically that black box explanation tools are preferred, and Kernel SHAP has an advantage over LIME of requiring little to no hyperparameter optimization. We also note that the gradient-based methods appear to be useful but can have unpredictable outcomes for low energy sources, possibly because the spectra contain orders of magnitude

more counts at the low end than at other portions of the spectrum. In addition, we demonstrate the utility of counterfactual explanations with a specific example.

## Explanation methods

The common notation we will use for the explanation methods is that we have a model $f$ that takes a spectrum $\mathbf{x} \in \mathbb{R}^N_{\geq 0}$ with $N$ bins as input, and outputs $C$ class scores $\mathbf{y} \in \mathbb{R}^C$. These scores are assumed to be the output logits before the softmax function that is typically used to normalize the outputs of models. The specific data point at issue (rather than the generic input) is $\mathbf{x}_0$, and bolding will be dropped when indexing individual vector elements.

Explanations will be denoted as $\phi \in \mathbb{R}^M$, where $M \leq N$ is the dimension of the explanation space. If $M < N$ then $\phi$ will be expanded proportionally to cover all $N$ input dimensions, and this is done in all figures showing explanations.

### Saliency mapping

Saliency mapping [28] was first proposed in 2013 as a means of interpreting the output of the latest generation of deep neural networks with high achievement on the ImageNet problem [35]. This method gave researchers some of their first insights into the overall operation of deep and complex ANNs. Saliency mapping was first used for gamma-ray spectra in reference [18].

A saliency map is the gradient of the output (e.g., the class score $y_c$ for class $c$, before the final softmax) with respect to the input data $\mathbf{x}$, evaluated on the original input data $\mathbf{x}_0$:

$$\left(\phi^c_{\text{saliency}}\right)_i \quad \propto \left.\frac{\partial y_c}{\partial x_i}\right|_{\mathbf{x}_0}. \tag{1}$$

Typically, the absolute value of $\phi^c_{\text{sal}}$ is taken and scaled to the range $[0, 1]$, but we will examine the raw gradient here.

Saliency, of course, requires that the model be differentiable. Since our models are implemented in Tensorflow library [36], this gradient can be calculated automatically.

### Grad-CAM

Gradient-weighted Class Activation Mapping (Grad-CAM) was introduced as a way of visualizing which regions of an image contribute most strongly to a particular class score [30]. Grad-CAM requires that at least one layer of the model is convolutional, and it weights the gradients of the class score with respect to the final convolutional layer by the activations of that layer itself. Assuming $A_{ij}(\mathbf{x}_0)$ is the activation of filter $j$ (out of a total of $J$) at index $i$ (out of a total of $M$) for the last convolutional layer in the model, then the Grad-CAM explanation is defined to be

$$\left(\phi^c_{\text{Grad-CAM}}\right)^*_i \quad \propto \text{ReLU}\left[\sum_{j=0}^{J-1}\left(\sum_{i'=0}^{M-1}\left.\frac{\partial y_c}{\partial A_{i'j}}\right|_{\mathbf{x}_0}\right) \cdot A_{ij}(\mathbf{x}_0)\right]. \tag{2}$$

However, this formula led to unusual outputs with our model and data. The culprit seemed to be the wide dynamic range of a single spectrum—by averaging the gradient over the spectral dimension ($i'$) first, large gradients, especially near the low energy portion of the spectrum where there may be orders of magnitude more counts than higher energy regions, were overly influencing the resulting mean gradients. We observed that if those gradients were not

averaged but were instead first multiplied by the activations, then the activations would serve to suppress the spurious high gradients, resulting in more understandable outputs.

In other words, the modification to Grad-CAM that was suitable for our spectra was to remove the gradient averaging and leave only the sum over the filter dimension,

$$(\phi^c_{\text{Grad-CAM}})_i \quad \propto \text{ReLU}\left[ \sum_{j=0}^{J-1} \frac{\partial y_c}{\partial A_{ij}}\bigg|_{\mathbf{x}_0} \cdot A_{ij}(\mathbf{x}_0) \right]. \tag{3}$$

The need to modify Grad-CAM serves to underline how methods developed for image data, which in general have lower dynamic ranges than gamma-ray spectra, may not necessarily give sensible results when used without modification, and care must be taken to ensure they are working as expected in the new context.

## LIME

LIME was proposed in 2016 as a way of making more robust explanations by creating local linear (and thus interpretable) versions of the model [34]. The basic principle is to repeatedly perturb the input data randomly and evaluate the model on those perturbations, and then to build a low-dimensional linear model that, at least in the neighborhood around the input data, closely approximates the original model. By examining the intrinsically more interpretable linear model, one can identify the main features of the input data that influenced the model's output.

Open source code for LIME is available [37], but we wrote our own version. To implement LIME, one needs three main ingredients: a scheme for masking the input data, a kernel for measuring the "distance" between different inputs, and a method for determining the optimal number of linearized features.

Masking is performed by selecting a number $M \leq N$ simplified input dimensions, with the choice of $M$ chosen generally to speed up computation. Each mask is represented by a simplified input $\mathbf{z} \in \{0, 1\}^M$, and $\mathbf{z}$ is converted to a masked version of $\mathbf{x}_0$ using the mapping function

$$\mathbf{h}_{\mathbf{x}_0}(\mathbf{z}) : \{0, 1\}^M \mapsto \mathbb{R}^N_{\geq 0}. \tag{4}$$

This mapping function is shared by SHAP and will be described in Method for masking spectral regions for LIME and SHAP.

The kernel for LIME is

$$\pi_{\mathbf{x}_0}(\mathbf{z}) \quad = \exp(-D^2(\mathbf{x}_0, \mathbf{h}_{\mathbf{x}_0}(\mathbf{z}))/\sigma^2), \tag{5}$$

where $D(\mathbf{x}_1, \mathbf{x}_2)$ is a distance metric between two spectra, and $\sigma$ is a distance scale parameter. For spectral data, which contain integer counts, we chose a statistically motivated distance metric based on Poisson statistics, the Poisson deviance [38],

$$D^2(\mathbf{x}_1, \mathbf{x}_2) \quad \equiv 2\sum_{i=0}^{N-1}\left[ (x_2)_i - (x_1)_i + (x_1)_i \log\left(\frac{(x_1)_i}{(x_2)_i}\right) \right], \tag{6}$$

which has the constraint that $\mathbf{x}_2$ cannot have any elements that are zero unless the same element in $\mathbf{x}_1$ is also zero, in which case the logarithm term becomes zero. In the limit of large counts in all bins of $\mathbf{x}_1$ and $\mathbf{x}_2$, the Poisson deviance becomes the chi-squared statistic. Also, when the gross counts of $\mathbf{x}_1$ and $\mathbf{x}_2$ are equal, the Poisson deviance simplifies to be proportional to the Kullback-Leibler divergence [39] between the normalized spectra.

Under the masking scheme, one of the largest distances achieved by any spectrum will be when the elements of $\mathbf{z}$ are all zero. Therefore, we chose to use this distance as a scale in the

kernel, so

$$\sigma \quad \equiv \tilde{\sigma}\sqrt{D^2(\mathbf{x}_0, \mathbf{h}_{\mathbf{x}_0}(\mathbf{0}_M))}, \tag{7}$$

where $\mathbf{0}_M$ is a mask of all zeros and $\tilde{\sigma}$ is a scale parameter that is less dependent on the characteristics of $\mathbf{x}_0$. We chose a default value of $\tilde{\sigma} = 1.0$.

Finally, LIME finds an explainable model $g$ that locally approximates the behavior of the class score $y_c$ around $\mathbf{x}_0$ by minimizing the following objective function

$$L(y_c, g, \pi_{\mathbf{x}_0}) \quad = \sum_{\mathbf{z} \in \mathcal{Z}} \pi_{\mathbf{x}_0}(\mathbf{z})[y_c(\mathbf{h}_{\mathbf{x}_0}(\mathbf{z})) - g(\mathbf{z})]^2 + \Omega(g), \tag{8}$$

where $\mathcal{Z}$ is a random sample of masks from the space $\{0, 1\}^M$, $g(\mathbf{z})$ is a linear model that takes the simplified inputs as its arguments,

$$g(\mathbf{z}) \quad = \phi_0 + \phi_{\text{LIME}}^c \cdot \mathbf{z}, \tag{9}$$

and $\Omega(g)$ is a penalty on the number of nonzero coefficients of $g$.

There are multiple ways presented to optimize the number of nonzero coefficients of $g$, denoted $K$. We chose one of the methods in the LIME open source repository [37], that of solving Eq 8 using $\Omega(g) = 0$ and ridge regression, finding the dimensions with the largest coefficients by absolute value, and then solving for a second time using ridge regression with only those dimensions. In all of the results presented later, LIME was performed by choosing $K = 10$, a ridge regression parameter $\alpha = 1 \times 10^{-2}$, and generating 1,000 random masks.

## SHAP (Kernel SHAP)

SHAP is a method that connects the LIME approach with a more general game-theoretic interpretation, namely that the input data features are viewed as "players" in a cooperative game for which there is an average payoff for each feature's participation [29]. The name for these payoffs are Shapley values, and in SHAP they take the place of the LIME coefficients.

Open source code is available [40], but, as was the case with LIME, we wrote our own code that used the masking scheme described in Method for masking spectral regions for LIME and SHAP.

To be calculated exactly, SHAP requires evaluating the class score for all possible masks except the mask of all zeros and the mask of all ones (i.e., $2^M - 2$ masks in total). Fortunately, the Shapley values can be approximated using only a random subset of all possible masks. This approximation is called Kernel SHAP [29] and is found by recasting LIME through the following definitions:

$$\pi_{\mathbf{x}_0}(\mathbf{z}) \quad = \frac{M-1}{\binom{M}{\|\mathbf{z}\|_0} \|\mathbf{z}\|_0(M - \|\mathbf{z}\|_0)}, \quad \mathbf{z} \notin \{\mathbf{0}_M, \mathbf{1}_M\} \tag{10}$$

$$g(\mathbf{z}) \quad = \phi_0 + \phi_{\text{SHAP}}^c \cdot \mathbf{z} \tag{11}$$

$$\Omega(g) \quad = 0, \tag{12}$$

where $\|\mathbf{z}\|_0$ is the metric that counts the nonzero elements of $\mathbf{z}$, and $\mathbf{0}_M$ and $\mathbf{1}_M$ are masks of all zeros and ones, respectively. Solving Eq 8 leads to the explanation $\phi$.

The resulting approximate Shapley coefficients end up in our experience to be similar in size and magnitude to the results of LIME, despite having virtually no hyperparameters to

define—the only hyperparameters are the number of simplified inputs, $M$, and the number of random mask samples in the sum in Eq 8.

## Method for masking spectral regions for LIME and SHAP

As has been described, to implement LIME and SHAP, a scheme to mask portions of the original input data is needed. The mapping from mask to spectrum is summarized by the mapping function $\mathbf{h}_{\mathbf{x}_0}(\mathbf{z})$. For the equivalent of image superpixels, so that computation efficiency of LIME and SHAP can be increased, the user is allowed to choose $M$ contiguous spectral regions, such that $M$ is less than or equal to the number of spectral bins $N$. The regions then consist of $M$ sets of $\approx N/M$ adjacent bins (with rounding to the nearest integer). For a given mask $\mathbf{z} \in \{0, 1\}^M$, the spectral regions to be masked out are indicated by where the values of $\mathbf{z}$ are zero.

For image data, both LIME and SHAP replace masked image superpixels with a representative mean color. In LIME, the choice is to mask using the mean of each color channel over the entire image, typically leading to a shade of gray. In SHAP, numerous options are used, including in-filling with the average color of neighboring regions. (These choices are not explained in the original papers [29, 34], but have been implemented in the corresponding open source codes [37, 40].) The purpose of "graying out" superpixels is to convert the original image regions into regions that do not have any features that result in any significant class activations, thus allowing the method to effectively disable the effects from the masked superpixels.

For gamma-ray spectra, since spectral bin values can vary over orders of magnitude in a single spectrum, inserting the mean value of the spectrum into a masked region would be a poor choice, since doing so could lead to the injection of highly unusual spectral features with unpredictable class activations. Inserting the mean value of neighboring spectral bins would be more suitable and lead to more natural-looking spectra. Other options we considered are to linearly interpolate between the values on the boundary of the masked region, or to linearly interpolate in the logarithm of the values. In reference [19], which used SHAP, and reference [20], which used LIME in addition to SHAP, it is not clear in either case what type of scheme was used to disable spectral features.

We found an additional way of masking gamma-ray spectra that resulted in the most stable class activations. The basic idea is to insert appropriately scaled portions of the mean background spectral shape, making the masked regions locally resemble the mean background and thus be unlikely to activate any non-background classes. The mean background spectral shape $\bar{\mathbf{x}}$ is calculated from all the spectra labeled as background in the training set, by taking the simple mean of all of the spectral bins. For a measured spectrum $\mathbf{x} \in \mathbb{R}_{\geq 0}^N$, each contiguous masked region was replaced by the same region of the mean spectrum, scaled to the gross counts of $\mathbf{x}$ within that region. In other words, if the simplified inputs $\mathbf{z}$ indicate a spectral region from $i_0$ to $i_1$ (inclusive) is to be masked, then the $i$th element of the masked spectrum, assuming $i$ is within that region, is

$$(h_{\mathbf{x}}(\mathbf{z}))_i = \left( \frac{\sum_{i=i_0}^{i_1} x_i}{\sum_{i=i_0}^{i_1} \bar{x}_i} \right) \bar{x}_i. \tag{13}$$

Among the implications here are that when all elements of $\mathbf{z}$ are zero, this method results in the entire mean background spectrum scaled by the gross counts of $\mathbf{x}$:

$$\mathbf{h}_{\mathbf{x}}(\mathbf{0}_M) = \left( \frac{\sum_{i=0}^{N-1} x_i}{\sum_{i=0}^{N-1} \bar{x}_i} \right) \bar{\mathbf{x}}. \tag{14}$$

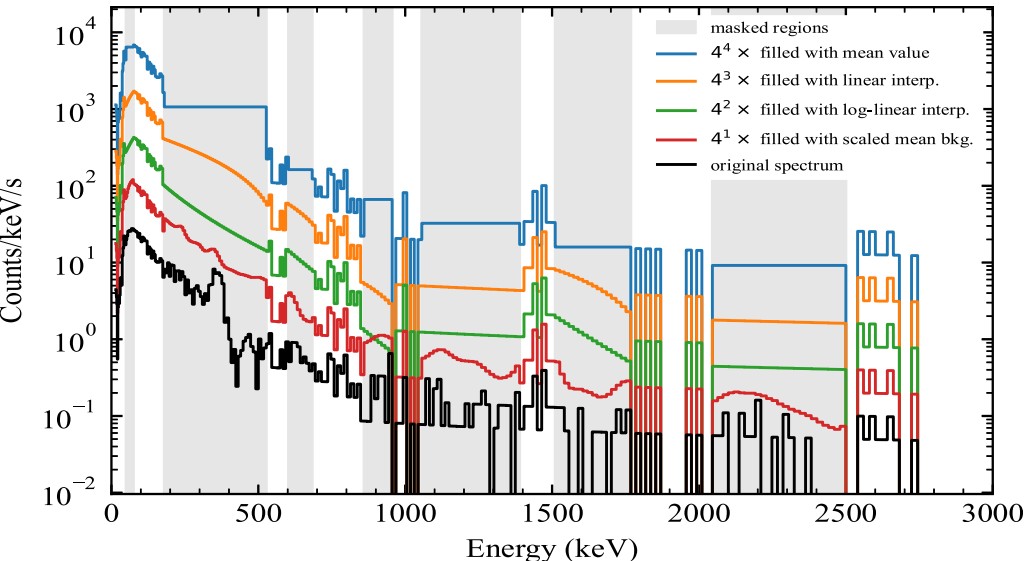

**Fig 1. Spectrum masking methods.** Example of masking a gamma-ray spectrum (black) using different approaches mentioned in the text. The spectrum contains an $^{131}$I anomaly, whose main photopeak can be seen at 364 keV.

An analogous process would not be suitable for most image datasets, since the visual field usually changes a much greater amount than the mean spectral shape of gamma-ray data.

An example of applying these different masking methods to a spectrum is shown in Fig 1. In that example, the region including the $^{131}$I photopeak at 364 keV is masked out, so one can compare how the different methods handle that region.

To see not just what the different masking methods look like, but how they may affect an algorithm, the class scores for the model described in Model for detection and identification were observed for multiple spectra. For a spectrum that was labeled background, the method that resulted in the most narrow and symmetric distribution of scores around the original score was the mean background method. An example of one such class score is shown in the top plot in Fig 2). In addition, a spectrum that contained $^{131}$I was examined with masking (in fact, the same spectrum shown in Fig 1). For that spectrum, because it contained a source, all of the score distributions were more variable than in the background case. However, the preferred masking method still results in the tightest distributions, as can be observed in the bottom of Fig 2.

## Counterfactual explanations

When a model gives significant confidence levels for multiple sources, a technique to explain why one class was chosen over the others may provide significant value. An explanation need not be calculated only for the best class score; indeed any class score can be used, so in these cases multiple explanations can be generated, one for each of the most likely sources. However, since sources that could be easily confused by the model tend to have features in the same region(s) of the spectrum, the multiple explanations on their own may not be helpful to a user when adjudicating the results. Instead, one may want to generate a *counterfactual* explanation for why one class was deemed more likely than the other, and such an explanation may reveal the features that most strongly distinguish between the two classes [41].

To examine the contrast between two explanations $\phi_1$ and $\phi_2$, we considered them each as two real-valued vectors in a Euclidean space (e.g., $\mathbb{R}^M$ in the case of LIME and Kernel SHAP).

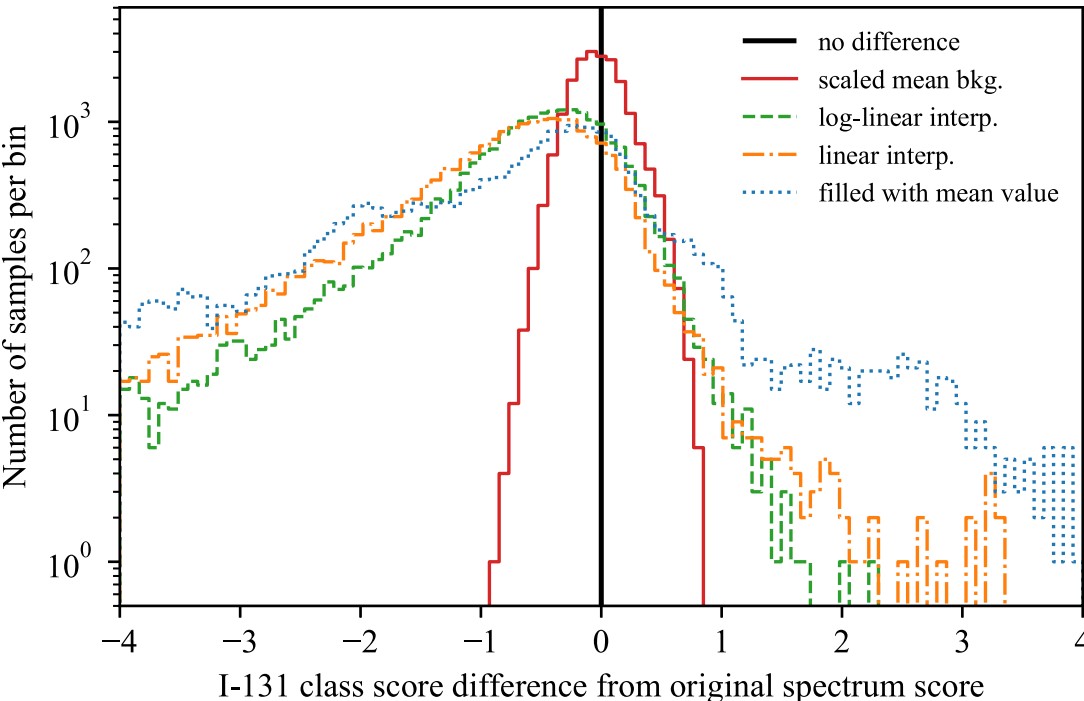

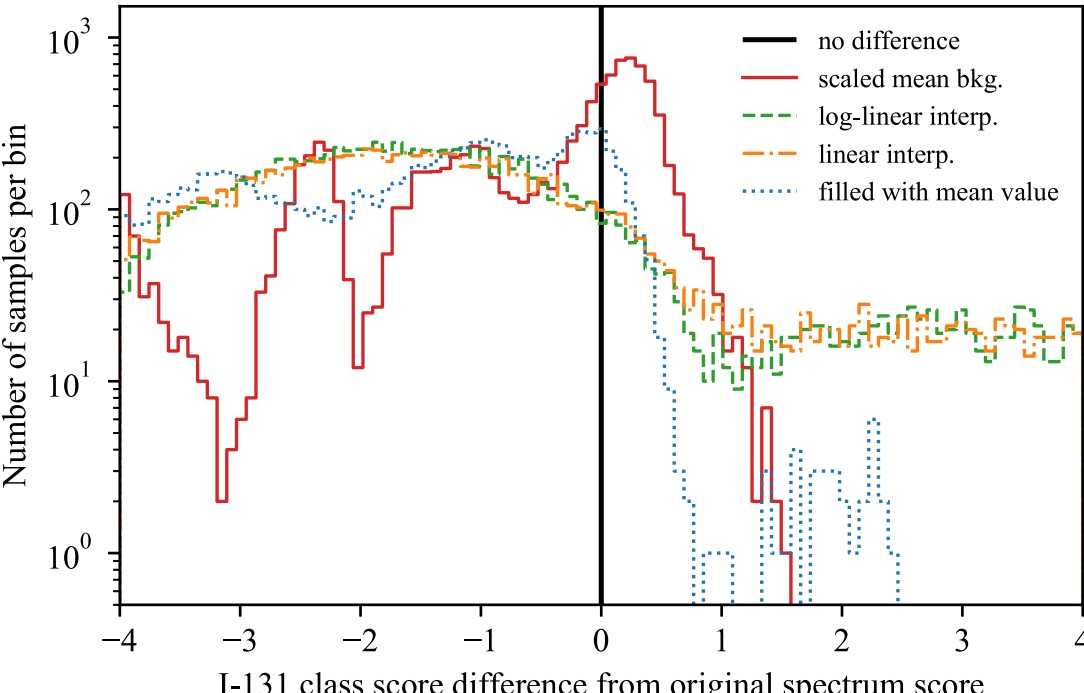

**Fig 2. The effect of the masking methods on model class scores.** Examples of the distribution of the model's class scores for [131]I using different masking methods. At top are the class scores for [131]I for a background spectrum, and the bottom shows the same quantities, but for the spectrum shown in Fig 1, which contains [131]I. Each histogram contains the results of 20,000 random masks.

Since the classes are both likely, the two explanations may be largely colinear, so the largest contrast between the explanations is in how they are orthogonal. In particular, assuming $\phi_1$ to be the higher confidence explanation and $\phi_2$ the weaker one, we propose using the negative of the orthogonal projection of $\phi_2$ onto $\phi_1$ as a technique for explaining why class 1 should be chosen over class 2, i.e.,

$$\phi_{1\perp2} \quad \equiv \left(\frac{\phi_2 \cdot \phi_1}{\phi_1 \cdot \phi_1}\right)\phi_1 - \phi_2. \tag{15}$$

An example of such a counterfactual explanation will be shown in Results.

For the explanation methods that use random masking (LIME and Kernel SHAP), we found the counterfactual explanations most clear when $\phi_1$ and $\phi_2$ were calculated using the same set of random masks, by, e.g., supplying both methods with the same random seed.

## Data and model

Here we will describe the data set used to train machine-learning models for spectral anomaly detection and identification, and the optimized model and its performance. This model and these data will be used in Explanation methods to demonstrate the explanation methods.

### The dataset and its preparation

The data used were derived from the RADAI dataset [42–44]. The RADAI dataset is the result of modeling an urban area to generate realistic radiological backgrounds, including $^{40}$K, $^{238}$U, and $^{232}$Th series emission from buildings, road materials, and soil; $^{214}$Pb and $^{214}$Bi emission from horizontal surfaces during rain events; cosmic-induced gamma-ray backgrounds; scattering and attenuation from people and vehicle clutter; and other effects.

In addition to backgrounds, 24 sources of various kinds were present in many encounters throughout the dataset. These sources emerged from point-like sources and comprised a number of types—naturally occurring radioactive material (NORM) anomalies, medical sources, industrial sources, and nuclear material—and contained various levels of shielding. The 24 source anomaly types (non-background) used in this study are listed in Table 1.

For each isotope, the dataset contained at least 30 "runs" of 3–5-minutes in duration that consisted of encounters with the source in different physical locations of the city blocks. The runs were divided into training runs and validation runs, with 60% of the runs used for training, 20% for validation and 20% used for testing. The list-mode data from each run were integrated into one-second long spectra, and source- and background-tagged events were binned separately.

The spectra were binned into 256 nonlinear bins, spaced from 15 to 3000 keV. In order to approximate the energy resolution as a function of energy, a square-root binning scheme was

**Table 1. Anomaly types in the RADAI dataset.**

| Category | Isotopes |
|---|---|
| NORM | $^{40}$K, $^{232}$Th series, $^{226}$Ra series |
| Medical | $^{57}$Co, $^{18}$F, $^{99m}$Tc, $^{131}$I, $^{201}$Tl, $^{67}$Cu, $^{90}$Sr, $^{177}$Lu, $^{133}$Xe |
| Industrial | $^{60}$Co, $^{137}$Cs, $^{133}$Ba, $^{192}$Ir |
| Nuclear | Depleted U, Natural U, Refined U, Low Enriched U, Highly Enriched U, Fuel-Grade Pu, Weapons-Grade Pu, $^{241}$Am |

used, i.e., the $j$-th bin edge ($j = 0...256$ inclusive) was

$$E_j \quad = \left( \sqrt{15} + \frac{\sqrt{3000} - \sqrt{15}}{256} j \right)^2 \text{ keV}. \tag{16}$$

This scheme is similar to the nonlinear scheme used in reference [22]. The intention behind this nonlinear binning scheme here and in that reference is to reduce the dimensionality of the spectral data without losing information, since no natural spectral features can appear that are at a higher resolution than the detector energy resolution.

Training, validation, and testing data were augmented in the following way. Binomial downsampling was applied to the *background* list-mode events using a probability randomly selected between 0.5 and 1.0. Binomial downsampling was applied to the *source* list-mode events with a probability randomly chosen logarithmically between $1 \times 10^{-6}$ and $1 \times 10^{0}$. Augmented spectra were retained so long as the number of source events to total events in the downsampled spectrum was above 1% (unless background spectra were being generated, in which case all spectra were accepted). In this way 20,000 spectra for each source type (including background) were created for the training set, and 5,000 spectra of each kind for the validation and testing sets. The tools for data generation and augmentation that were used are in the open source `radai` repository [45].

## Model for detection and identification

For demonstration purposes in this paper, we will use a simple ANN and apply it to the classification problem for spectra, i.e., determining whether a spectrum is background or whether it contains any of the 24 kinds of anomalies, and if so, which one. The models used were inspired by the architectures used by [12, 18, 19], and consisted of two 1-D convolutional layers, each followed by a max pooling layer, followed by two fully connected layers each followed by dropout, and a finally fully connected layer activated by a softmax to the final 25 outputs. As used in references [11, 12], the input spectra were first preprocessed by dividing by the maximum bin value to scale all of the bin values to be between 0 and 1, i.e., $\mathbf{x} \rightarrow \mathbf{x}/\max(\mathbf{x})$. Unlike reference [12], the outputs were not fractional abundances but categorical probabilities, so the training was done with one-hot representations of the labels. The models were optimized by minimizing sparse cross entropy, and the Adam optimizer [46] was used, with a batch size of 1024, a learning rate of $1 \times 10^{-4}$, and early stopping after 20 epochs if the validation-set loss has not decreased. The model was implemented in Keras [47] and Tensorflow [36] versions 2.11.0.

The best model was found by a random search over the parameter set given in Table 2, and the full details of the model architecture are listed in Table 3. The architecture results in a model with a total of 674,315 trainable weights.

**Table 2. Model hyperparameter search.**

| Hyperparameter | Search space | Best value |
|---|---|---|
| 1-D Conv. no. of filters | [10, 20, 30, 40, 50] | 30 |
| 1-D Conv. kernel width | odd numbers from 9–25 | 21 |
| Max pool size | [No pool, 2, 3, 4] | 2 |
| Max pool stride | [1, 2, 4] | 1 |
| Dropout rate | [0.3, 0.5, 0.7] | 0.5 |
| Fully connected nodes | [100, 200, 300, 500] | 100 |

**Table 3. Model architecture.**

| Layer type | Output dimensions | Parameters |
|---|---|---|
| Input | (256,) | — |
| 1-D Convolution | (236, 30) | filters = 30, kernel width = 21, stride = 1 |
| Activation | (236, 30) | ReLU |
| Max pool | (235, 30) | pool size = 2, stride = 1 |
| 1-D Convolution | (215, 30) | filters = 30, kernel width = 21, stride = 1 |
| Activation | (215, 30) | ReLU |
| Max pool | (214, 30) | pool size = 2, stride = 1 |
| Flatten | (6420,) | — |
| Dropout | (6420,) | dropout rate = 0.5 |
| Fully connected | (100,) | — |
| Activation | (100,) | ReLU |
| Dropout | (100,) | dropout rate = 0.5 |
| Fully connected | (100,) | — |
| Activation | (100,) | ReLU |
| Dropout | (100,) | dropout rate = 0.5 |
| Fully connected | (25,) | — |
| Activation | (25,) | Softmax |

## Model performance

The model was used to assign a predicted class to each of the spectra in the testing set by taking the arg max of the predicted categorical probabilities. The overall accuracy of the model in identifying the 25 categories was 30.9%. The identification performance can be seen in the confusion matrix in Fig 3, which shows the predicted class label for each of the true class labels. The dataset was meant to be challenging to the algorithm, so a large number of spectra are identified as background. Some anomaly classes are more easily distinguished from background than others, such as $^{133}$Xe, which has strong emission at the lowest part of the spectrum. Other low energy sources, such as $^{241}$Am and $^{201}$Tl, also have better detection performance than other source types. Aside from detection performance, some anomaly types have poor identification performance, especially refined uranium (RefinedU), fuel-grade plutonium (FGPu), and weapons-grade plutonium (WGPu). For example, FGPu can be seen to have strong cross-talk with both $^{241}$Am and WGPu, which explains where the wrong identifications are primarily being made. Though not shown here, the confusion between these three classes persists even at high SNR values. Other notable features in the confusion matrix are the significant crosstalk groups formed by the different forms of uranium (depleted uranium (DU), refined uranium, low enriched uranium (LEU)) and the different plutonium-related anomalies ($^{241}$Am, FGPu, and WGPu).

## Results

To examine the different explanation methods, we applied them to spectra from the testing set. We began by examining spectra where the algorithm both correctly detected a non-background class and correctly identified the source. One of the most straightforward cases is $^{137}$Cs, which consists of a single photopeak at 662 keV with an associated continuum of down-scattered events. Fig 4 shows the four explanation methods applied to a spectrum that truly contained $^{137}$Cs and that the model identified as $^{137}$Cs with 98.7% confidence. For convenience and to highlight the most obvious non-background features, the mean background scaled to

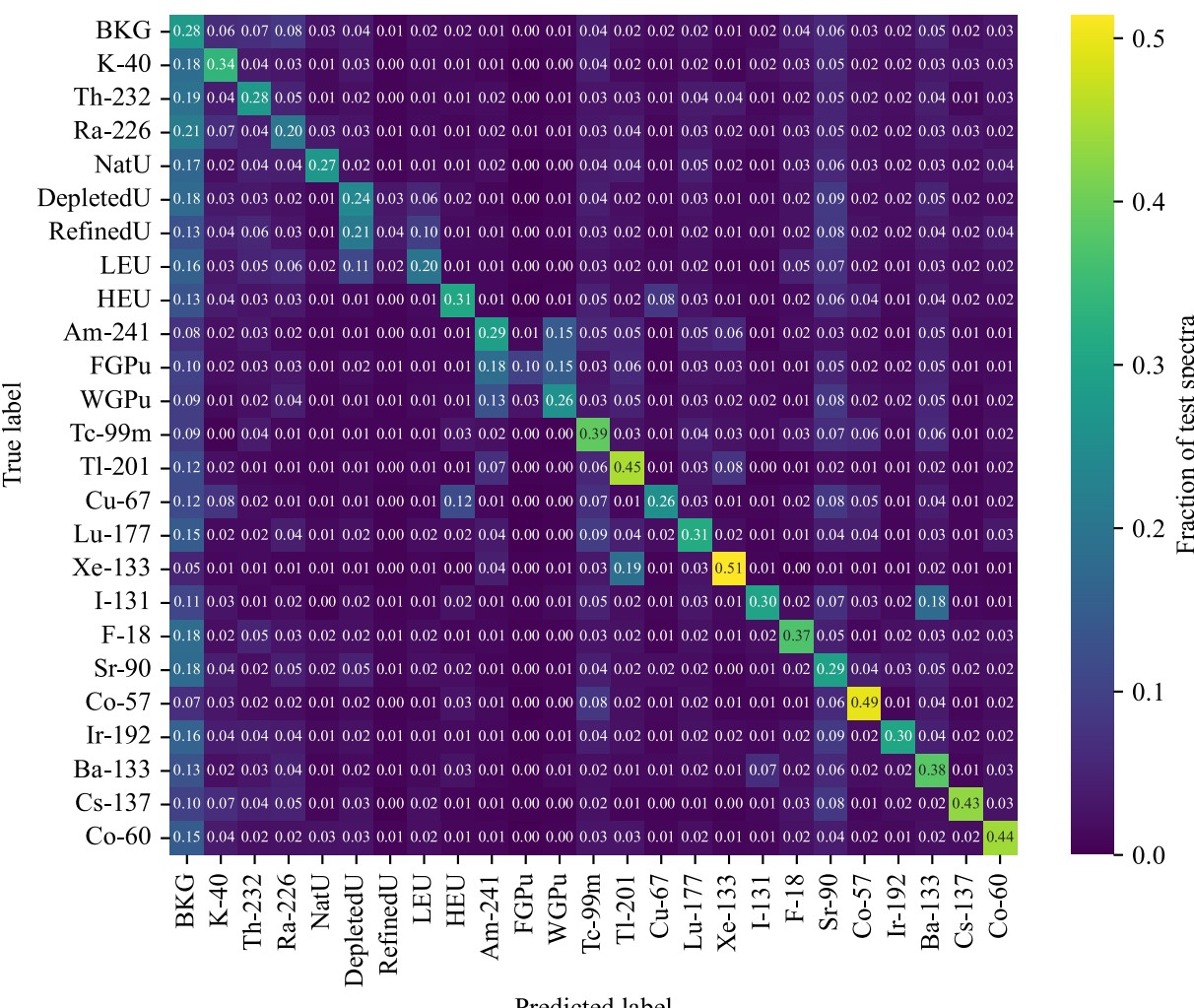

**Fig 3. Confusion matrix of the model.** The confusion matrix for the 25 spectrum categories obtained by evaluating the model on the testing set.

match the gross counts of the spectrum is overlaid on each plot, and the deviance residuals from this scaled mean background (i.e., signed square root of the individual terms in the sum in Eq 6, which are approximately the standardized residuals for Poisson data) from this simple fit are plotted beneath. In addition, the spectrum of only those events known to be from the source (because each event is tagged by its origin in the RADAI dataset) is also plotted to qualitatively verify whether the explanations are able to identify the actual regions of highest importance. In the case of $^{137}$Cs, the four methods do indeed identify the region around the 662 keV photopeak as the most important, which can be seen as also being the region with the highest signal-to-background and where there is a distinct positive residual from the mean background shape. Of note are that LIME and Kernel SHAP return nearly identical results both in distribution and magnitude. (Here and in what follows, LIME and Kernel SHAP use a simplified input size of $M = 64$, or "superpixels" consisting of groups of every four spectral bins, and LIME is calculated using $K = 10$.).

The low energy source $^{201}$Tl was also used to compare the four methods (Fig 5). In this case, the results diverged significantly between the methods. The two gradient-based methods failed

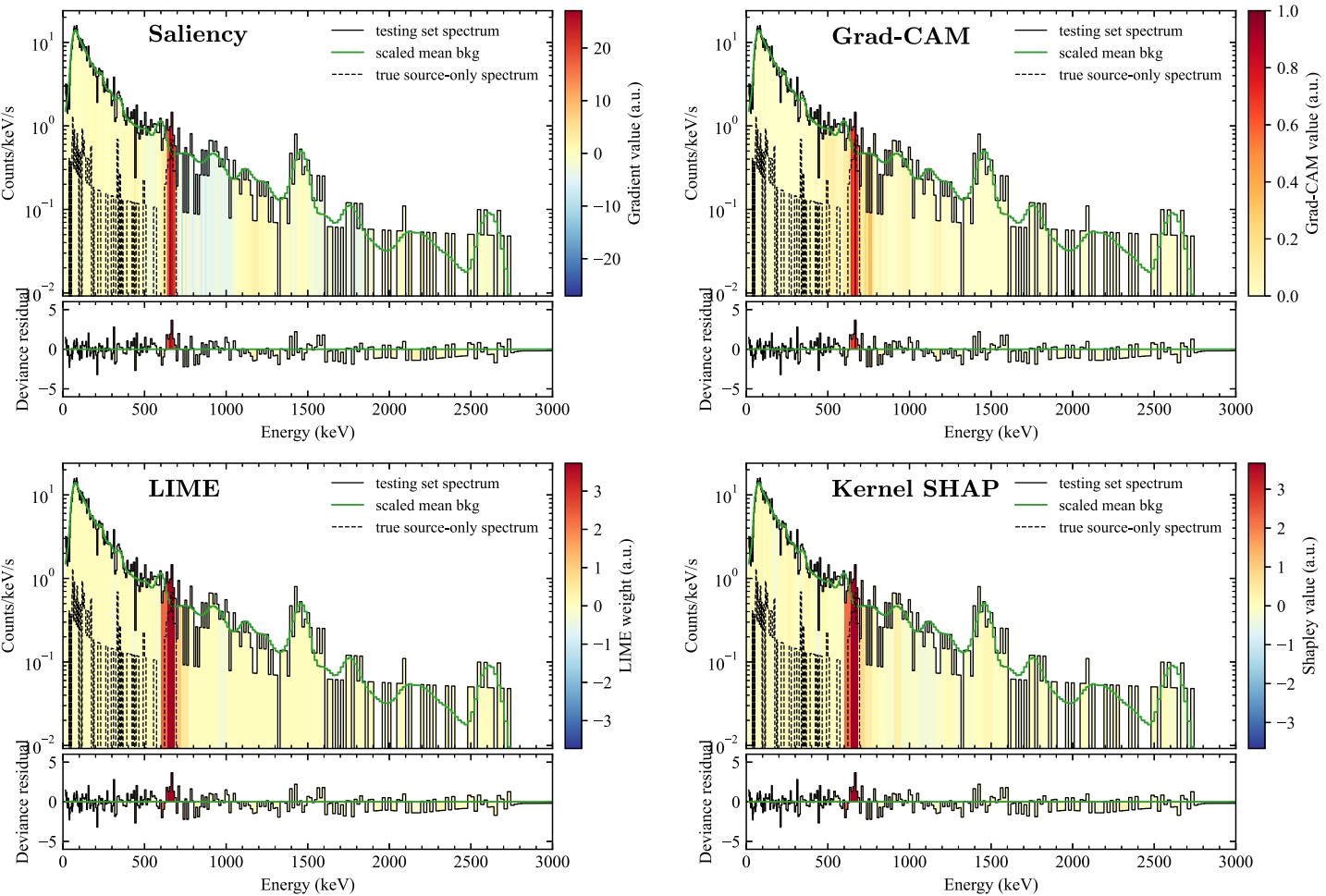

**Fig 4. Comparison of explanation methods for a spectrum with $^{137}$Cs.** A comparison of the four explanation methods considered here for a spectrum containing a medium energy source, $^{137}$Cs. For each spectrum, the scaled mean background spectrum is shown for reference, and the deviance residual from the scaled mean background is shown beneath.

to correctly identify the region where the source was the strongest, with the saliency map having large gradients throughout the spectrum and not just in the region 40–80 keV, where most of the source events are. In addition, Grad-CAM highlighted only the first several bins, below 40 keV. These problems may be due to how non-uniform the data in each spectrum can be—here the nonzero elements of the spectrum vary over more than two orders of magnitude, with the highest counts at the low end of the spectrum. It may be that the gradients in the low energy region can dominate in unpredictable ways because of this. Indeed, issues with the uniformity of the gradient had to be dealt with earlier in modifying Grad-CAM (Grad-CAM), since its original outputs were often nonsensical when tested with spectral data. Some initial testing with models trained using logarithmic normalization of the data (as was done in [18]) seems to mitigate some of these effects, at least for saliency mapping.

Meanwhile, LIME and Kernel SHAP again produced outcomes in close agreement with each other, and both highlight the 60–80 keV region, which is at the higher end of the region with the most source counts. This emphasis on the 60–80 keV region may be because of other source types (e.g., $^{241}$Am and the types of plutonium) that have strong features at 60 keV and below, giving that region higher specificity to this particular isotope.

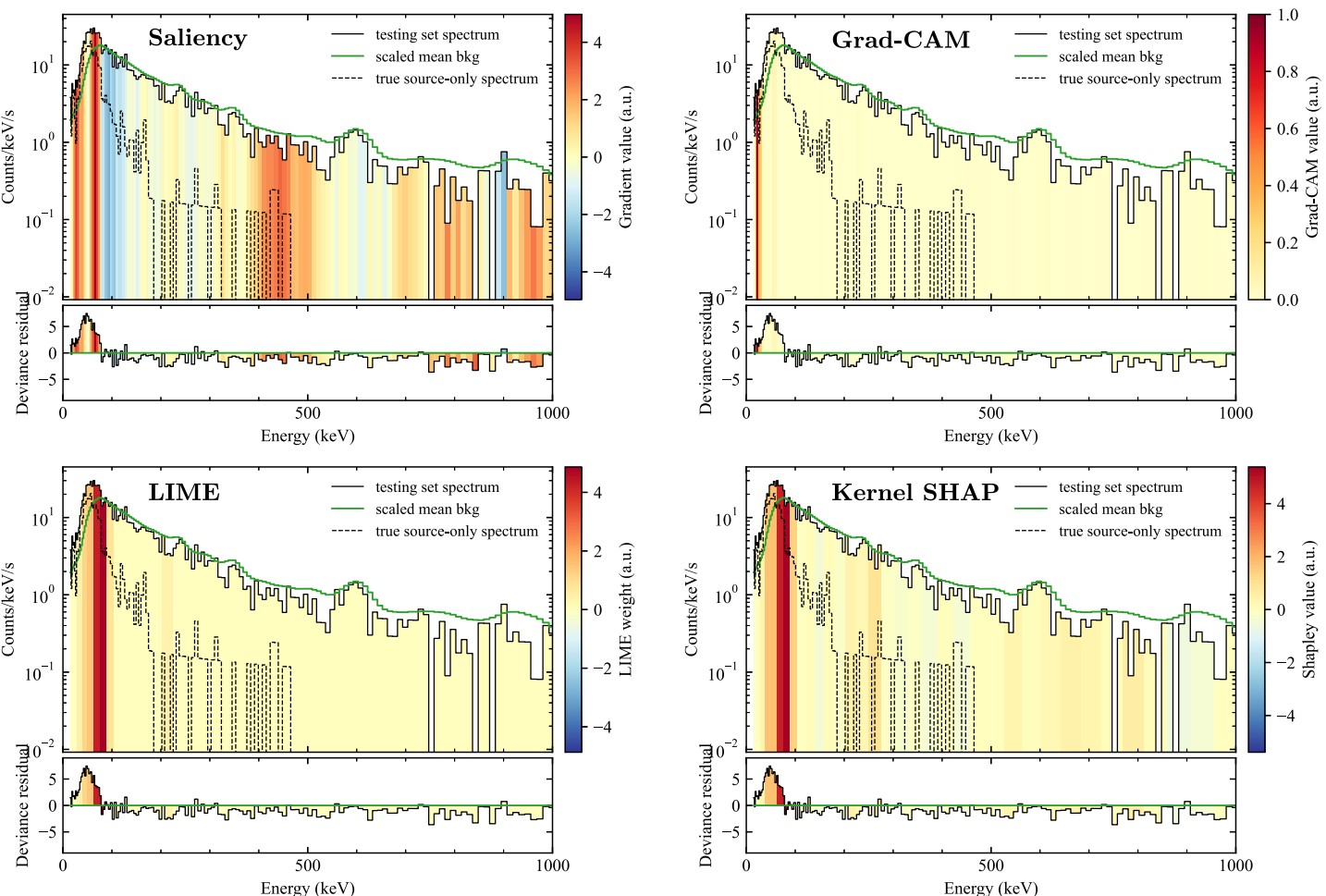

**Fig 5. Comparison of explanation methods for a spectrum with $^{201}$Tl.** A comparison of the four explanation methods considered here for a spectrum with a low energy source, $^{201}$Tl. For clarity, the spectrum is shown only up to 1000 keV.

To demonstrate the usefulness of the explanation methods in a more complex situation, we examined sources that can take on significantly different spectral shapes due to different amounts of shielding. One of the clearest and most important examples is weapons-grade plutonium (WGPu), which, when unshielded has a dominant 60 keV line from $^{241}$Am, but when shielded the signature is primarily characterized by a complex of lines around 350–400 keV. Fig 6 shows the results of saliency mapping and Kernel SHAP for two such spectra. In both cases, the methods generally highlight the correct regions, but saliency mapping has more extraneous nonzero values throughout the rest of the spectrum, even in areas where there are few to no source counts, showing that saliency mapping could potentially cause confusion for the end user.

As a final demonstration of the usefulness of the methods, we considered counterfactual explanations for when the model gives significant confidence levels for multiple sources, as described in Counterfactual explanations. Fig 7 shows one such comparison, where a spectrum containing $^{131}$I was assigned a confidence of 88.5%, and $^{133}$Ba was given a 9.0% confidence. The explanations for the two isotopes using Kernel SHAP reveal qualitatively similar distributions, though with slightly different overall scales. For both isotopes, the region

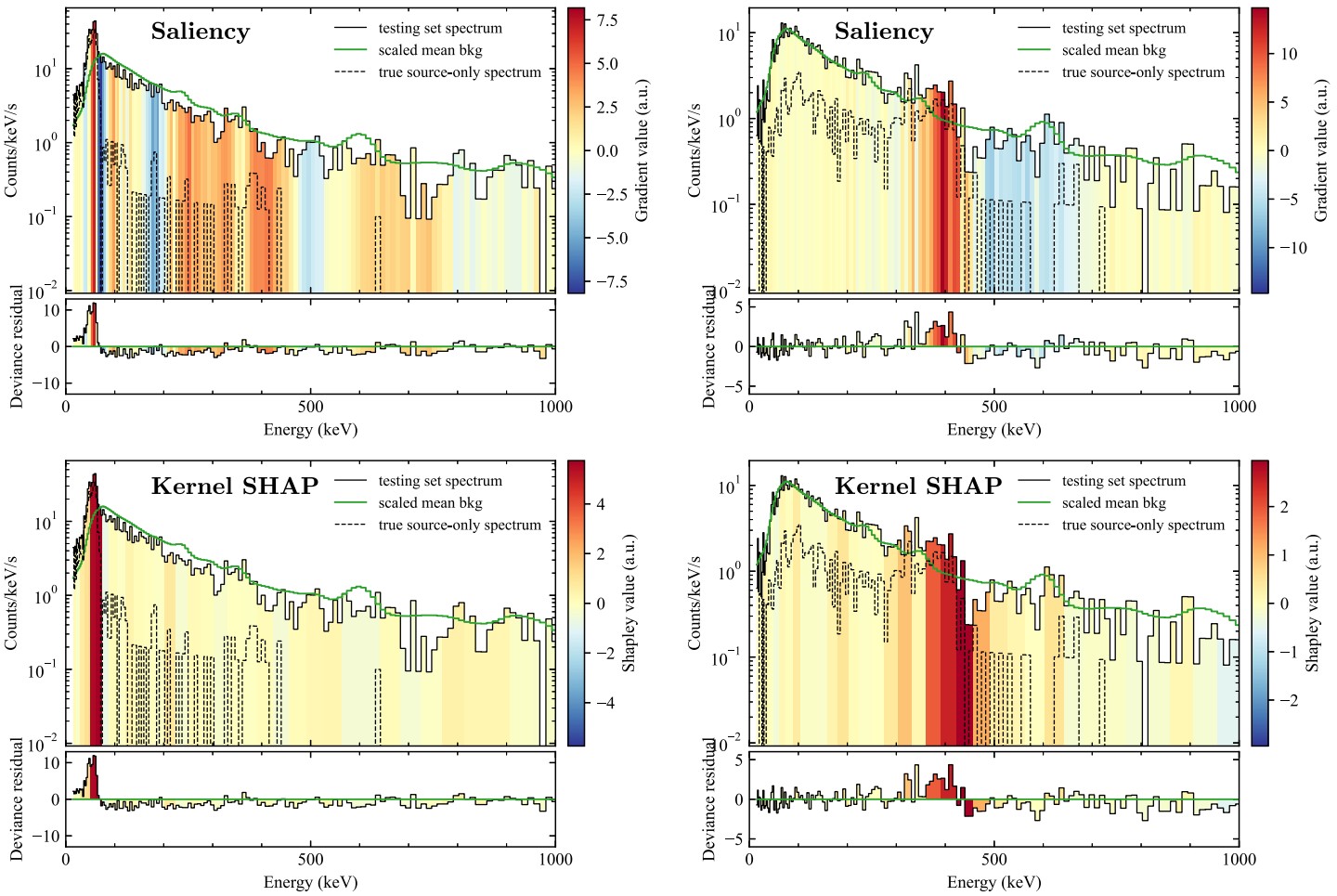

**Fig 6. Explanations for shielded and unshielded versions of the same source.** A comparison of saliency (top) and Kernel SHAP (bottom) for unshielded (left) and shielded (right) weapons-grade plutonium sources (WGPu). For clarity, the spectrum is shown only up to 1000 keV.

around 330–420 keV is found to be important, which comports with the intuition that the strongest photopeaks of the two isotopes are their most distinguishing features (364 keV for $^{131}$I and 356 keV for $^{133}$Ba). However, the energy resolution is too large to distinguish those two photopeaks from each other, so it would be useful for adjudication purposes to find out how the model has been able to find other means of contrast between the two isotopes. The bottom plot of Fig 7 shows the counterfactual explanation $-\phi_{(I-131) \perp (Ba-133)}$, i.e., the most important features for choosing $^{131}$I over $^{133}$Ba. This explanation is illustrative, since it reveals a moderate importance of the region of approximately 600–650 keV, which is where $^{131}$I has a significant photopeak (637 keV) but $^{133}$Ba does not. Additionally, the slight excess of events in the region 250–300 keV are assigned negative values, indicating that their existence provides support for the presence of $^{133}$Ba. This result makes sense because $^{133}$Ba has two significant lines in that region (276 keV with 7% branching and 303 keV with 18% branching), while $^{131}$I has fewer relevant photopeaks there (284 keV with 6% branching) [48].

A second example of a counterfactual explanation is shown in Fig 8, where the model incorrectly identifies the isotope. The spectrum actually contains $^{133}$Xe, but it has been

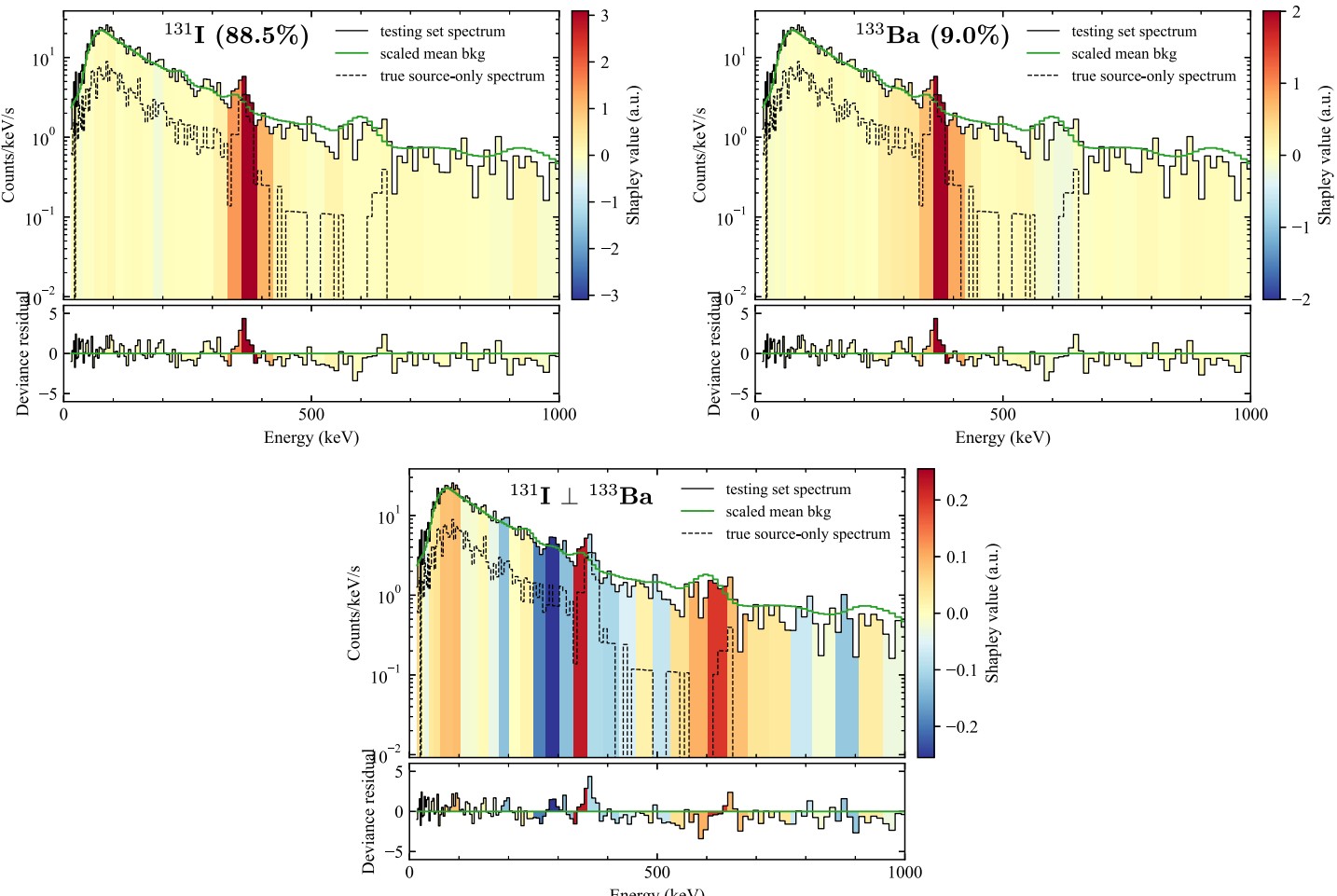

**Fig 7. Example of a counterfactual explanation.** Kernel SHAP used for explanations of the class score for $^{131}$I (top left), which is the true anomaly contained by the spectrum, and an explanation of the class score for $^{133}$Ba, to which the model also gave significant confidence. The bottom middle is a plot of Kernel SHAP used to generate a counterfactual explanation for why $^{131}$I was correctly selected over $^{133}$Ba.

assigned a confidence of only 15.6%, while $^{201}$Tl has been assigned 84.3%. In the range 50–100 keV, which the explanations indicate are the most important region for both isotopes, $^{133}$Xe has only one prominent line at 81.0 keV (36.9%), while $^{201}$Tl has several, ranging from 68.9 to 82.5 keV (90.7% total among them) [48]. Thus $^{133}$Xe's emission should be generally higher than that from $^{201}$Tl, however the true source-only spectrum indicates that the actual anomalous emission is peaked around 50–60 keV, likely due to shielding of the source.

The counterfactual explanation $-\phi_{(Tl-201) \perp (Xe-133)}$, shown in the bottom of Fig 8, gives the most important features in reaching the incorrect determination of $^{201}$Tl instead of the correct one. The counterfactual explanation is strongly negative in the region approximately 90-100 keV, indicating that the model saw less emission in that region than expected for $^{133}$Xe, and that is why the confidence is lower for it. Examining a case like this may indicate to a researcher that more development is needed to correctly distinguish between these two isotopes when there is shielding, potentially guiding the collection of more training data or the pursuit of other model architectures.

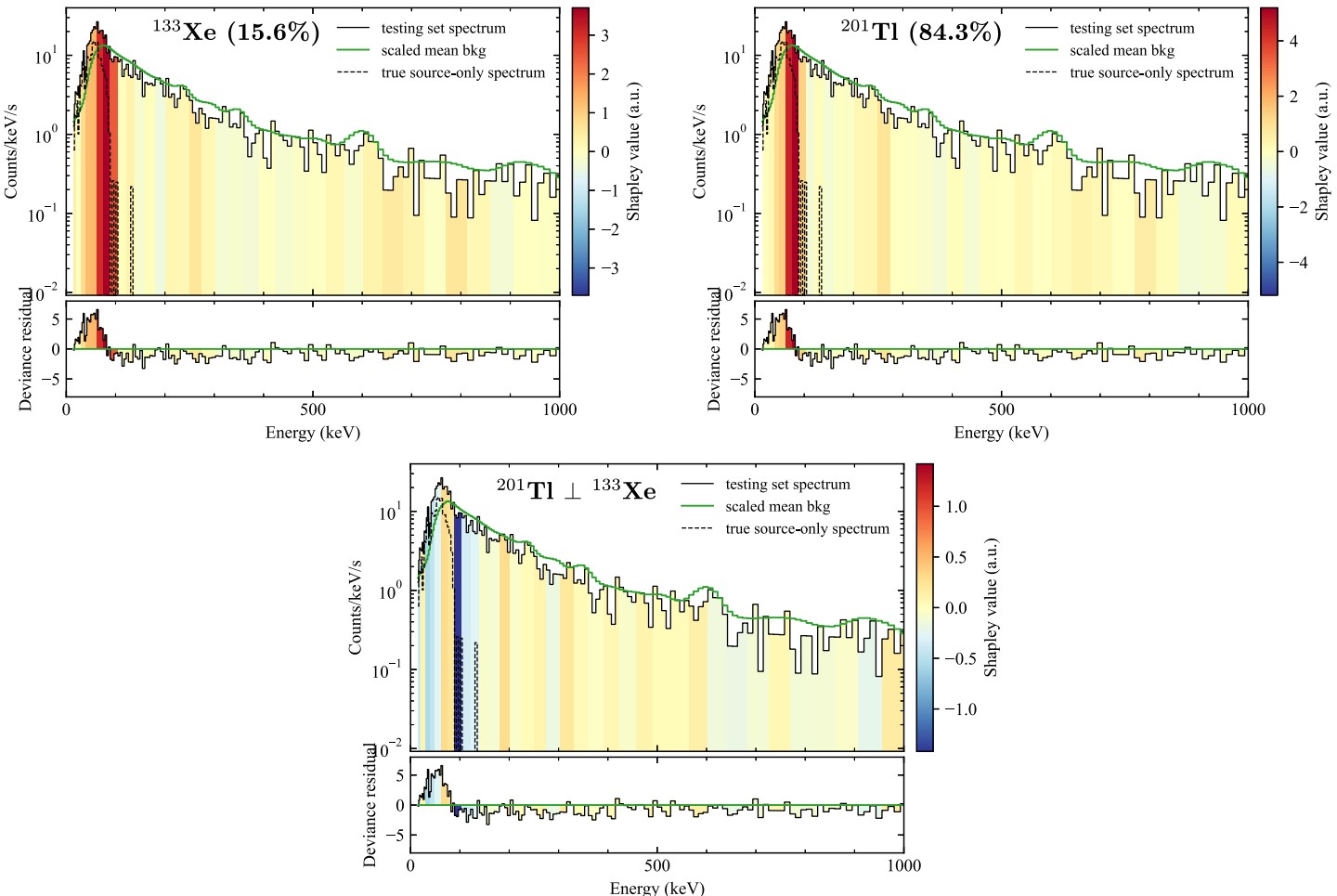

**Fig 8. Example of a counterfactual explanation when incorrect class was identified.** Kernel SHAP used for explanations of the class score for [133]Xe (top left), which is the true source present, but the model incorrectly assigned a higher confidence to [201]Tl (top right). The bottom middle is a plot of the counterfactual explanation for why the incorrect [201]Tl was selected instead of the correct [133]Xe.

## Discussion

Methods to explain the outputs of machine-learning models provide context that the model's predictions on their own do not. In gamma-ray spectroscopy, and in the particular application of anomaly detection and identification in urban search, these explanations can provide insights into which regions of the spectrum are most important. As others have noted, these regions tend to be those where there are important features such as isotope-specific photo-peaks [18–20].

However, there are additional complexities in applying explanation tools to spectral data. The first is that the gradient-based methods we explored (saliency mapping and Grad-CAM) seem to become distorted at the lower energies. We suspect that the gradients become dominated by their values at low energies, which are orders of magnitude higher than other regions of the spectrum, and that if these tools are to be useful going forward then some adequate regularization must be found. To deal with the low energy effects, we had to modify Grad-CAM to suppress the effects of low-energy gradients, although the result was not entirely successful. These problems with gradients at the lowest energies may be behind the unusual result seen in

the saliency map for $^{241}$Am in reference [18], where the clearly visible 60 keV line was not highlighted by the method, while the higher energy photopeak regions of $^{137}$Cs and $^{60}$Co were correctly found. The $^{241}$Am result was especially surprising given that the model's performance on $^{241}$Am is nearly 100% perfect according to the confusion matrix.

An additional lesson is that some explanation tools require masking of the input data features in a way that is appropriate for the data. The default masking behavior implemented in open source versions of the black box methods like LIME and SHAP may work well for images and other data but not for spectra. For example, inserting the mean value of the data into the masked regions might work for images, but creates highly distorted gamma-ray spectra. Linearly interpolating masked regions leads to more stable class scores for randomly masked spectra, and we found that inserting scaled regions of the mean background shape worked even better. The existing applications of LIME [20] and SHAP [19, 20] in the literature do not clearly specify how masking was done, although their results support the usefulness of SHAP.

Another result of this paper is that LIME and Kernel SHAP are found to give comparable and nearly identical results. This coincidence is not surprising given that SHAP is a generalization of LIME [29]. These results are at odds with reference [20], where LIME is found to return different (and useless) results compared to SHAP. However, in that paper, in addition to the masking scheme, the hyperparameters used for LIME were not stated, such as which method was used for determining the optimal number of nonzero linear coefficients $K$, and which parameters were used when performing any relevant Lasso or ridge regressions. Without knowing these choices, one cannot conclude whether LIME was truly useless for those data, or whether it had not been properly tuned.

The necessity for tuning LIME, including specifying a distance metric, a masking scheme, and a method for finding $K$, gives it more drawbacks than Kernel SHAP. Kernel SHAP shares with LIME a need for a masking scheme, a method for dividing a spectrum into a desired number ($M$) of "superpixels," and an adequate number of random masks to generate. (Care should be taken to use the Kernel SHAP approach and not the full Shapley value calculation, which requires evaluating the model for *all* possible masks.) Therefore, of all the methods we examined, based on the findings of this work we recommend the general adoption of Kernel SHAP over the other methods.

Lastly, we found that a user of complex spectral models could benefit not just from explanations, but also from explanations of contrast between other likely outcomes. We propose the orthogonal projection of Kernel SHAP explanations as a simple and effective way to derive such an explanation. Second-order analytical tools, like these explanations of class contrast, have not been explored in this application space but are a natural extension of them.

## Acknowledgments

The authors thank Brian J. Quiter for providing helpful suggestions for this manuscript.

## Author Contributions

**Conceptualization:** Mark S. Bandstra, James M. Ghawaly, Jr.

**Data curation:** Joseph C. Curtis, James M. Ghawaly, Jr.

**Formal analysis:** Mark S. Bandstra.

**Funding acquisition:** Tenzing H. Y. Joshi.

**Investigation:** Mark S. Bandstra, A. Chandler Jones.

**Methodology:** Mark S. Bandstra, James M. Ghawaly, Jr.

**Project administration:** Tenzing H. Y. Joshi.

**Software:** Mark S. Bandstra, Joseph C. Curtis.

**Supervision:** Tenzing H. Y. Joshi.

**Visualization:** Mark S. Bandstra.

**Writing – original draft:** Mark S. Bandstra.

**Writing – review & editing:** Mark S. Bandstra, Tenzing H. Y. Joshi.

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
