## [Decision Letter · Decision Letter 0]

9 Feb 2023

PONE-D-22-35119Explaining machine-learning models for gamma-ray detection and identificationPLOS ONE

Dear Dr. Bandstra,

Thank you for submitting your manuscript to PLOS ONE. After careful consideration, we feel that it has merit but does not fully meet PLOS ONE’s publication criteria as it currently stands. Therefore, we invite you to submit a revised version of the manuscript that addresses the points raised during the review process.

We look forward to receiving your revised manuscript.

Kind regards,

Mohammad Amin Fraiwan

Academic Editor

PLOS ONE

Journal Requirements:

2.We note that the grant information you provided in the ‘Funding Information’ and ‘Financial Disclosure’ sections do not match. 

"The project was funded by the U.S. Department of Energy, National Nuclear Security Administration, Office of Defense Nuclear Nonproliferation Research and Development."

"This work was performed under the auspices of the U.S. Department of Energy by Lawrence Berkeley National Laboratory (LBNL) under Contract DE-AC02-05CH11231.

The project was funded by the U.S. Department of Energy, National Nuclear Security Administration, Office of Defense Nuclear Nonproliferation Research and Development.

This  manuscript  has  been  authored  in  part  by  UT-Battelle,  LLC,  under  contract  DE-AC05-00OR22725  with  the US Department of Energy (DOE).

The US government retains and the publisher, by accepting the article for publication, acknowledges that the US government retains a nonexclusive, paid-up, irrevocable, worldwide license to publish or reproduce the published form of this manuscript, or allow others to do so, for US government purposes.

DOE will provide public access to these results of federally sponsored research in accordance with the DOE Public Access Plan (http://energy.gov/downloads/doe-public-access-plan).

Additional Editor Comments:

Kindly make sure that appropriate machine learning evaluation methods are used and reported. The paper will not be accepted without the proper evaluation of the proposed work.

Reviewers' comments:

Reviewer's Responses to Questions

**Comments to the Author**

1. Is the manuscript technically sound, and do the data support the conclusions?

Reviewer #1: Yes

Reviewer #2: Yes

Reviewer #3: Yes

2. Has the statistical analysis been performed appropriately and rigorously? 

Reviewer #1: No

Reviewer #2: Yes

Reviewer #3: I Don't Know

3. Have the authors made all data underlying the findings in their manuscript fully available?

Reviewer #1: Yes

Reviewer #2: No

Reviewer #3: Yes

4. Is the manuscript presented in an intelligible fashion and written in standard English?

Reviewer #1: Yes

Reviewer #2: Yes

Reviewer #3: Yes

5. Review Comments to the Author

Reviewer #1: This is an interesting paper about XAI applications on a gamma-ray spectroscopy study. However, few important points should addressed before considering it for publication.

Major issues:

1) A workflow of the procedure is needed to better understand the work done. A graphical schematization of the ANN used would also be appreciable.

2) It's not very clear to me how you set the network parameters. In my opinion a cross validation procedure is needed to rule out possible neural network overfitting issues.

3) Once the cross validation procedure is implemented it would be better to put the average performance with uncertainties inside the confusion matrix

4) The confusion matrix is fine since you have many classes but it would be good to provide some overall performance through AUC and accuracy as well.

Minor issues

1) What version of keras and tansor flow did you use? Through which framework?

Reviewer #2: The authors present work at improving Machine Learning tools used for analyzing gamma ray spectra. There are many applications for wanting to take a gamma ray spectra (NaI (Tl) detectors are the example here) and turn it into a list of what isotopes are contributing. This can be a mess, since there's so many things that might be in there (especially in the case of spent fuel), so machine learning techniques to automate this analysis would be quite useful. The authors extend and improve methods for improving things, especially on the very important "but _why_ do the algorithms like this solution?" front, rewriting and documenting some algorithms that showed promise in other papers but which where not as well documented as one would like.

The paper was very well written: every time I found myself asking "hmm - what's that? Why does that work?" I didn't have to wait very long to encounter a good explanation, backed up with data and figures. This paper will be a very handy reference not only for people working on ML in gamma ray spectroscopy, but also in other applications.

The authors say that the code and datasets will be available upon publication at publication, which is great! But it's not there yet: gitlab returns a 404: so the "no" to question #3 above is a qualified one: at this moment it's a "no", but it would be a "yes" if the authors follow through. I would have loved to poke around in it as part of this review.

Reviewer #3: The manuscript is structured well and written in a clear and intelligible manner. Authors investigate several explanation tools to analyze the basis of decisions, made by the ML model, and conclude that LIME and SHAP techniques are the most reliable between the covered techniques. The manuscript adds value and would be useful for the community, however I have some minor questions that I believe would further improve the quality of the paper

1- The architecture used for modeling the data is inspired by references, 12, 18 and 19. Have authors tried other architectures or this is the only one? Given the model agnostic nature of most explanation techniques, would be informative if the authors reported their observation, on two different architectures. Given the availability and effectiveness of AutoML models, this can be done in a short time.

2- Authors discuss cases from higher a and lower ends of the energy spectrum along with a more complex case with overlapping signature. I personally think, including an example of misclassification by the model, also has value to demonstrate how the model comes up with the wrong decision.

3- A normalized confusion matrix in lieu of, or accompanying figure 3 would make it easier to understand model performance.

6. PLOS authors have the option to publish the peer review history of their article (what does this mean?). If published, this will include your full peer review and any attached files.

Reviewer #1: No

Reviewer #2: **Yes**

Reviewer #3: No

---

## [Author Response · Author response to Decision Letter 0]

2 May 2023

Please see our responses in the attached file response_v6.pdf

---

## [Decision Letter · Decision Letter 1]

24 May 2023

Explaining machine-learning models for gamma-ray detection and identification

PONE-D-22-35119R1

Dear Dr. Bandstra,

We’re pleased to inform you that your manuscript has been judged scientifically suitable for publication and will be formally accepted for publication once it meets all outstanding technical requirements.

Kind regards,

Mohammad Amin Fraiwan

Academic Editor

PLOS ONE

Additional Editor Comments (optional):

Reviewers' comments:

Reviewer's Responses to Questions

**Comments to the Author**

1. If the authors have adequately addressed your comments raised in a previous round of review and you feel that this manuscript is now acceptable for publication, you may indicate that here to bypass the “Comments to the Author” section, enter your conflict of interest statement in the “Confidential to Editor” section, and submit your "Accept" recommendation.

Reviewer #1: All comments have been addressed

2. Is the manuscript technically sound, and do the data support the conclusions?

Reviewer #1: Yes

3. Has the statistical analysis been performed appropriately and rigorously? 

Reviewer #1: Yes

4. Have the authors made all data underlying the findings in their manuscript fully available?

Reviewer #1: Yes

5. Is the manuscript presented in an intelligible fashion and written in standard English?

Reviewer #1: Yes

6. Review Comments to the Author

Reviewer #1: I thank the authors for responding comprehensively to my comments. The work is well written and potentially very interesting.

7. PLOS authors have the option to publish the peer review history of their article (what does this mean?). If published, this will include your full peer review and any attached files.

Reviewer #1: **Yes: **Alfonso Monaco

---

## [Editor Report · Acceptance letter]

8 Jun 2023

PONE-D-22-35119R1 

Explaining machine-learning models for gamma-ray detection and identification 

Dear Dr. Bandstra:

I'm pleased to inform you that your manuscript has been deemed suitable for publication in PLOS ONE. Congratulations! Your manuscript is now with our production department. 

Kind regards, 

on behalf of

Dr. Mohammad Amin Fraiwan 

Academic Editor

PLOS ONE